# Assessing eligibility for lung cancer screening using parsimonious ensemble machine learning models: A development and validation study

**Thomas Callender** [1] *, **Fergus Imrie** [2], **Bogdan Cebere** [3], **Nora Pashayan** [4], **Neal Navani** [1], **Mihaela van der Schaar** [3,5,6‡], **Sam M. Janes** [1‡]

1 Department of Respiratory Medicine, University College London, London, United Kingdom, 2 Department of Electrical and Computer Engineering, University of California, Los Angeles, California, United States of America, 3 Department of Applied Mathematics and Theoretical Physics, University of Cambridge, Cambridge, United Kingdom, 4 Department of Applied Health Research, University College London, London, United Kingdom, 5 Cambridge Centre for AI in Medicine, University of Cambridge, Cambridge, United Kingdom, 6 Alan Turing Institute, London, United Kingdom

‡ These authors are joint senior authors on this work.
* t.callender@ucl.ac.uk

## Abstract

### Background

Risk-based screening for lung cancer is currently being considered in several countries; however, the optimal approach to determine eligibility remains unclear. Ensemble machine learning could support the development of highly parsimonious prediction models that maintain the performance of more complex models while maximising simplicity and generalisability, supporting the widespread adoption of personalised screening. In this work, we aimed to develop and validate ensemble machine learning models to determine eligibility for risk-based lung cancer screening.

### Methods and findings

For model development, we used data from 216,714 ever-smokers recruited between 2006 and 2010 to the UK Biobank prospective cohort and 26,616 high-risk ever-smokers recruited between 2002 and 2004 to the control arm of the US National Lung Screening (NLST) randomised controlled trial. The NLST trial randomised high-risk smokers from 33 US centres with at least a 30 pack-year smoking history and fewer than 15 quit-years to annual CT or chest radiography screening for lung cancer. We externally validated our models among 49,593 participants in the chest radiography arm and all 80,659 ever-smoking participants in the US Prostate, Lung, Colorectal and Ovarian (PLCO) Screening Trial. The PLCO trial, recruiting from 1993 to 2001, analysed the impact of chest radiography or no chest radiography for lung cancer screening. We primarily validated in the PLCO chest radiography arm such that we could benchmark against comparator models developed within the PLCO control arm. Models were developed to predict the risk of 2 outcomes within 5 years from

**Data Availability Statement:** To facilitate use of the UCL models, we have developed a website and have made the models themselves available: https://github.com/callta/lung-cancer-models. The underlying code for AutoPrognosis is available from https://github.com/vanderschaarlab/

AutoPrognosis. UK Biobank, NLST, and PLCO data were used on license (references 68073, NLST-806 and PLCO-801, respectively). These data are subject to material transfer agreements and cannot be shared directly. However, researchers can apply for these data from the UK Biobank (https://www.ukbiobank.ac.uk/) and the US National Institutes of Health (PLCO: https://cdas.cancer.gov/plco/; NLST: https://cdas.cancer.gov/nlst/).

**Funding:** This work was supported by the Wellcome Trust (222890/Z/21/Z to TC), the National Science Foundation (1722516 to FI and MvdS), the Medical Research Council (MR/T02481X/1 to NN and MR/W025051/1 to SMJ) and Cancer Research UK (EDDCPGM\100002 to SMJ). The funders had no role in study design, data collection and analysis, decision to publish, or preparation of the manuscript.

**Competing interests:** I have read the journal's policy and the authors of this manuscript have the following competing interests: NN reports honoraria for non-promotional educational talks, conference support or advisory boards from Amgen, Astra Zeneca, Boehringer Ingelheim, Bristol Myers Squibb, Guardant Health, Janssen, Lilly, Merck Sharp & Dohme, Olympus, OncLive, PeerVoice, Pfizer, and Takeda. SMJ has received fees for advisory board membership in the last three years from Astra-Zeneca, Bard1 Lifescience, and Johnson and Johnson. He has received grant income from Owlstone and GRAIL Inc. He has received assistance with travel to an academic meeting from Cheisi. TC and SMJ are founders of, and own stock in, Mortimer Health.

**Abbreviations:** AUC, area under the receiver operating curve; CT, computed tomography; E/O, expected/observed; LCRAT, Lung Cancer Risk Assessment Tool; LCDRAT, Lung Cancer Death Risk Assessment Tool; LDCT, low-dose computed tomography; LLP, Liverpool Lung Project; MICE, multiple imputation by chained equation; NLST, National Lung Screening Trial; PLCO, Prostate, Lung, Colorectal and Ovarian Cancer Screening Trial; USPSTF, US Preventive Services Taskforce.

baseline: diagnosis of lung cancer and death from lung cancer. We assessed model discrimination (area under the receiver operating curve, AUC), calibration (calibration curves and expected/observed ratio), overall performance (Brier scores), and net benefit with decision curve analysis.

Models predicting lung cancer death (UCL-D) and incidence (UCL-I) using 3 variables—age, smoking duration, and pack-years—achieved or exceeded parity in discrimination, overall performance, and net benefit with comparators currently in use, despite requiring only one-quarter of the predictors. In external validation in the PLCO trial, UCL-D had an AUC of 0.803 (95% CI: 0.783, 0.824) and was well calibrated with an expected/observed (E/O) ratio of 1.05 (95% CI: 0.95, 1.19). UCL-I had an AUC of 0.787 (95% CI: 0.771, 0.802), an E/O ratio of 1.0 (95% CI: 0.92, 1.07). The sensitivity of UCL-D was 85.5% and UCL-I was 83.9%, at 5-year risk thresholds of 0.68% and 1.17%, respectively, 7.9% and 6.2% higher than the USPSTF-2021 criteria at the same specificity. The main limitation of this study is that the models have not been validated outside of UK and US cohorts.

## Conclusions

We present parsimonious ensemble machine learning models to predict the risk of lung cancer in ever-smokers, demonstrating a novel approach that could simplify the implementation of risk-based lung cancer screening in multiple settings.

## Author summary

### Why was this study done?

- Screening and disease prevention programmes are increasingly bespoke; however, their simultaneous delivery at a population-scale presents considerable challenges.

- Lung cancer is the most common cause of cancer death worldwide, with poor survival in the absence of early detection.

- Screening for lung cancer among those at high-risk could reduce lung cancer-specific mortality by 20% to 24% among those screened, but the ideal way to determine if someone is high-risk remains uncertain and existing approaches are resource intensive.

### What did the researchers do and find?

- We used data from the UK Biobank and US National Lung Screening Trial to develop novel, parsimonious models, to simplify the prediction of lung cancer risk and selection to lung cancer screening programmes.

- Using ensemble machine learning and 3 predictors—age, smoking duration, and pack-years—we found our models achieved or exceeded parity in performance with leading comparators despite requiring one-third of the variables.

- Our models were externally validated in the US Prostate, Lung, Colorectal, and Ovarian (PLCO) Cancer Screening Trial and benchmarked against models that are either in use or have performed strongly in previous analyses.

**What do these findings mean?**

- Risk assessment for lung cancer screening can be simplified without reducing performance, potentially improving the uptake and effectiveness of national lung cancer screening programmes, and therefore contributing to reducing deaths from lung cancer.

- Future research should focus on the application of this modelling approach to other conditions such as cardiovascular disease, diabetes, and chronic kidney disease to support the implementation at scale of multiple concurrent risk-stratified prevention and early detection programmes for major causes of morbidity and mortality.

- This study has key limitations as it is based on past data from the US and UK, so prospective evaluation in different countries and regions should be considered.

## Introduction

Screening, early detection, and disease prevention programmes are increasingly bespoke, with risk prediction algorithms determining an individual's eligibility and management [1–3]. Such personalisation promises to improve the benefit-to-harm profile of such interventions and ultimately health outcomes [4–6]. However, the delivery of these programmes at a population scale requires 2 conditions of risk prediction models: that they generalise well to contexts where there are insufficient data for model development, retraining, or validation; and, that the trade-off between model complexity and implementation feasibility is considered.

Screening for lung cancer—the foremost cause of death from cancer worldwide [7]—with low-dose computed tomography (LDCT) has been associated with a 20% to 24% reduction in lung cancer-specific mortality among those at high risk and are screened [8,9]. However, the ideal method to identify those at high risk remains unresolved. The US Preventive Services Taskforce (USPSTF) recommends the use of dichotomous criteria—age 50 to 80, $\geq$20 pack-years smoked, and <15 quit-years for former smokers—to select screening participants [10]. Nevertheless, identifying individuals for lung cancer screening based on risk prediction models has been shown to have both better benefit-to-harm profiles and cost-effectiveness than using dichotomous risk factor-based criteria alone [11–14], leading to risk-model-based selection criteria in European lung cancer screening pilots [15].

To date, most externally validated prediction models for lung cancer have been developed in United States datasets [12,16–21], reflecting the relatively limited availability of suitable cohorts with long-term follow-up for prediction modelling. This implies that most global healthcare systems that implement risk-based lung cancer screening will use prediction models developed in a US population, often using variables such as ethnicity, whose categorisation varies between countries and individual datasets, and academic qualifications that differ both over time and between jurisdictions. In the United Kingdom, existing models have been shown to underperform in specific groups, such as the more socioeconomically deprived, where underestimation of risk could lead to a screening programme systematically widening health inequalities [22].

Furthermore, the risk models currently in use are a challenge to implement. In the UK, eligibility for lung cancer screening pilots is based on the PLCOm2012 and Liverpool Lung Project risk models, requiring 17 unique variables, few of which are routinely available [23].

Collecting these variables from an individual who is potentially eligible and explaining the results currently averages between 5 and 10 min. To determine the screening eligibility of 1 million people would therefore require between 48 and 95 full-time staff a year. In the UK, there are an estimated 6.8 million ever-smokers aged 55 to 74 who are potentially eligible for lung cancer screening, with another 500,000 turning 55 on average each year [24,25]. As lung cancer screening is just one of several risk-based programmes that are either in development or in use, in their current form, these assessments present a formidable obstacle to the effective implementation of national screening programmes.

In this study, we hypothesised that using ensemble machine learning with training data spanning different geographic regions, populations, and average risk levels, we could develop predictive models for lung cancer screening with a minimum number of features that has broad applicability. In so doing, we aimed to combine the simplicity of risk-factor-based criteria with the improved predictive performance of risk models, while maintaining generalisability to new settings.

## Methods

### Ethical statement

The University College London Research Ethics Committee gave ethical approval for this study (reference: 19131/001).

### Data sources and study population

**Development and internal validation datasets.** For model development, we first used data on 216,714 ever-smokers without a prior history of lung cancer from the UK Biobank [26] before creating a multicountry dataset that combined UK Biobank and US National Lung Screening Trial (NLST) [8] data ($n$ = 26,616) (Fig 1; participant flow diagrams in Figs A and B in S1 Appendix). The UK Biobank is a large prospective cohort recruited between 2006 and 2010 from 22 British centres that combines phenotypical data with ongoing linkage to hospital and registry data [27]. During this timeframe, the UK has not had a systematic screening programme for lung cancer. The NLST was a randomised controlled trial of lung cancer screening comparing computed tomography (CT) against chest radiography in 33 US centres between 2002 and 2004 with follow-up through 2009 [28]. Participation in the NLST was restricted to those considered at high risk of developing lung cancer: a 30 pack-year smoking history and, if a former smoker, to have quit within 15 years of enrolment [28].

We selected the NLST because it is geographically distinct, includes a higher risk cohort, and has greater ethnic diversity than the UK Biobank. By combining NLST data with the UK Biobank, which by contrast is known to represent a cohort with lower mortality risks than the UK general population [29], our prediction models would be trained on a wider range of participants, with potentially improved model performance.

**External validation datasets.** For model validation, we used data from 40,593 ever-smokers without a prior history of lung cancer from the chest radiography arm of the US Prostate, Lung, Colorectal and Ovarian Cancer Screening Trial (PLCO) [30] trial (Fig C in S1 Appendix). This allowed benchmarking against comparator models that were developed in the control arm of the PLCO trial. Chest radiography was found to have no impact on lung cancer mortality, nor a statistically significant impact on lung cancer incidence [30]. In secondary analyses presented in S1 Appendix, we report model performance in both arms of the full PLCO dataset together ($n$ = 80,659).

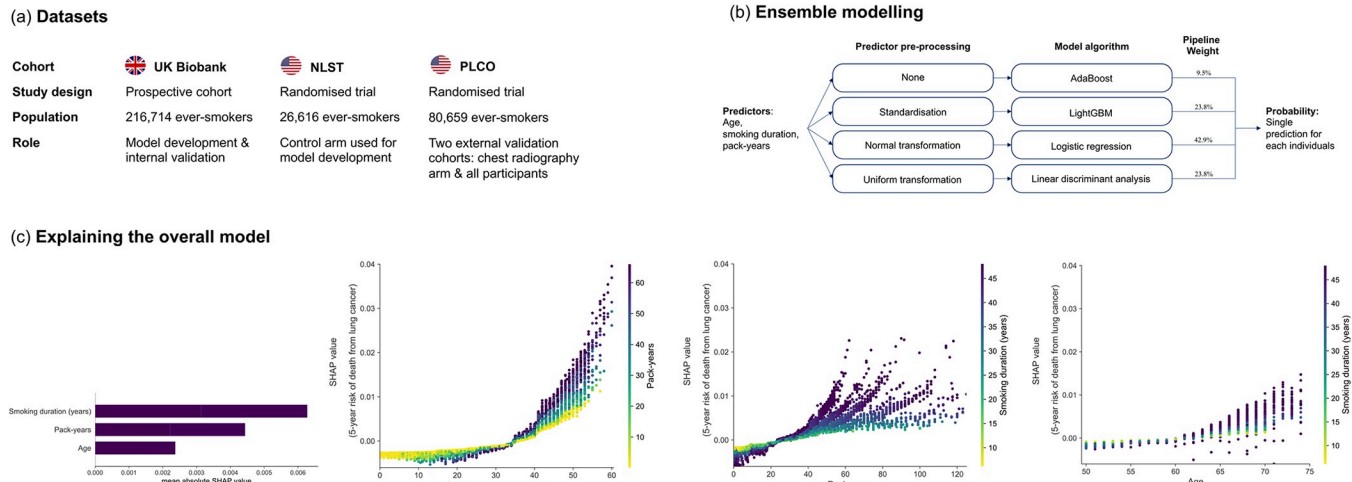

**Fig 1. Developing the UCL models to determine lung cancer screening eligibility.** A multicountry dataset comprising the UK Biobank and NLST was used to develop new models before external validation in the PLCO Trial chest radiography arm (allowing benchmark comparison with existing models developed in the PLCO control arm) and the full PLCO cohort (a). The ensemble modelling approach involves optimising individual modelling pipelines before combining their results as a single prediction for each individual. (b) Shows details of the UCL-D model, including the weights attributed to each pipeline in generating a single prediction for the five-year risk of lung cancer for any individual. (c) Shows the contribution of different variables to overall predictions as well as interactions between predictors, analysed using Shapely Additive Explanations (SHAP) on the UK Biobank [35]. The first subfigure in (c) shows that smoking duration was the most important variable when making predictions of an individual's risk of dying from lung cancer, followed by pack-years smoked, and finally age. The 3 subsequent dependence subplots show the relationship between the predictor (x-axis) against the outcome (y-axis)—the importance of knowing that predictor value when making a prediction. The vertical dispersion shows the degree of interaction effects present, while the colour corresponds to a second variable. The plots show that smoking for less than approximately 35 years had relatively little impact on model predictions, with a steep inflection and increasing interaction between smoking duration and pack-years after this point. This relationship between smoking duration and pack-years mirrors that seen in the previous subfigure, with duration trumping quantity of cigarettes smoked unless both are high. In other words, those individuals who smoke for short periods of time have a lower predicted risk, even if they smoke relatively large quantities. This reflects our understanding of lung biology and the ability of the lung to repair itself if an individual stops smoking [56]. Lastly among subfigures of (c), we see that age has relatively limited impact on the model under the age of 60. NLST, National Lung Screening Trial; PLCO, Prostate, Lung, Colorectal and Ovarian Cancer Screening Trial.

## Missing data

We used multiple imputation by chained equations (MICE) with predictive mean matching to generate imputed development and validation datasets [31]. We generated 10 imputed sets of the UK Biobank, based on an average missingness among candidate predictors in the UK Biobank of 11%. As missingness was <5% for all relevant variables in the PLCO and NLST, we created 5 imputed PLCO and NLST datasets. See Table A and Figs D–F in S1 Appendix for further details.

## Outcomes

We developed models to predict the absolute cumulative risk of 2 outcomes within 5 years from baseline: diagnosis of lung cancer and death from lung cancer. Lung cancer status and primary cause of death in the UK Biobank were determined by linked national cancer registry and Office for National Statistics data [26]. In the NLST and PLCO, primary cause of death was confirmed by independent review of death certificates [28,30]. In the NLST, lung cancer diagnoses were ascertained through medical record abstraction, and in the PLCO, through mailings and telephone calls to participants [8,30].

## Model development

We developed ensembles of machine learning pipelines using AutoPrognosis, open-source automated machine learning software [32,33]. In this analysis, AutoPrognosis was used to

optimise pipelines consisting of a variable preprocessing step followed by model selection and training. These optimised pipelines were subsequently combined and a single prediction for any individual generated by a weighted combination of the predictions made by each of the 4 pipelines independently, with weighting by Bayesian model averaging (Fig 1) [32]. We trialled model algorithms including logistic regression, random forests, and Gradient Boosting approaches (see subsection Model Development in S1 Appendix for further details). Throughout, pipelines were trained and selected to maximise model discrimination, measured with the area under the receiver operating curve (AUC).

## Model explanation

We used the Kernel Shapely Additive Explanations (SHAP) [34] algorithm for model explanation and analysis of predictor interactions (Fig 1). Kernel SHAP is a permutation-based method theoretically based in coalitional game theory. In summary, each variable is passed to a model one-by-one, with the change in predictions that occurs attributed to this model [35,36]. Further details are available in subsection Variable Importance and Interaction in S1 Appendix.

## Variable selection

For pragmatic reasons, we considered candidate predictors from the UK Biobank that were also present in the NLST and PLCO (Table B in S1 Appendix). We settled on our final list of predictors based on the literature, domain expertise, variable distributions, generalisability to multiple settings, and model discrimination in the UK Biobank. This led to the development of full models. Reviewing feature importance, we found that age, smoking duration, and pack-years were driving predictions, leading us to develop models using these 3 predictors (Fig G in S1 Appendix).

## Statistical analysis

We considered a model's overall performance with the Brier score [37], discrimination with the AUC, calibration with calibration curves and the ratio of expected-to-observed cases, and clinical usefulness with decision curve analysis [38]. Calibration curves were calculated by splitting individuals into 10 risk deciles based on their predicted risk before compared predicted probability against observed risk, the latter calculated using a Kaplan–Meier model. For a measure of clinical utility, we considered the net benefit of models across a range of risk thresholds [38]. We compared model discrimination with a two-tailed bootstrap test using the methods of Hanley and MacNeil, modified by Robin and colleagues [39,40]. To determine potential risk thresholds for our models, we used a fixed population strategy, comparing the number of individuals eligible for screening in the entire PLCO external validation dataset using the USPSTF-2021 criteria.

In both internal and external validation, we generated 1,000 bootstrap resamples with replacement for all analyses; central estimates and 95% confidence intervals were calculated with the percentile method. We used optimism-corrected metrics for internal validation. All analyses were conducted with R [41] and Python [42].

## Model comparisons

For benchmark comparisons, we compared our new models to the USPSTF-2021 criteria (age 50 to 80, ≥20 pack-year smoking history, and quit within the last 15 years if a former smoker) [10], as well as existing risk models that are either in use (PLCOm2012 [18] and Liverpool

Lung Project (LLP) version 2 [43]) or have been externally validated and consistently shown to outperform other risk models (the Lung Cancer Death Risk Assessment Tool [LCDRAT] and Lung Cancer Risk Assessment Tool [LCRAT] [19]) (Table C in S1 Appendix) [13,22,44,45]. All comparator models predict the five-year risk of death (LCDRAT) or developing lung cancer (LCRAT, LLP) except for the PLCOm2012 that predicts the six-year risk of lung cancer occurrence. A third, recalibrated, version of LLP has been developed. Because it is not currently in use, we present full comparative analyses in the Appendix but note that in using the same predictors and coefficients as LLP version 2, its discrimination is equivalent. Further, we also compared against Cox models developed using the same dataset (see Methods in S1 Appendix) and the constrained versions of the LCDRAT, LCRAT, and PLCOm2012 models.

All variables were available for comparator models except the LLP. For the LLP, in the UK Biobank, data were not available for age at which a family member developed lung cancer. Following ten Haaf and colleagues [44], and reflecting UK lung cancer epidemiology [46], we assumed that all with a family history of lung cancer were aged over 60. In the PLCO dataset, asbestos exposure and prior history of pneumonia were not available and were set to zero. We used the lcmodels package in R to calculate predictions for the PLCOm2012, LCRAT, and LCDRAT models [47].

## Results

The descriptive characteristics of the UK Biobank and NLST development datasets and the PLCO external validation dataset are presented in Table 1. Characteristics by outcome are presented in Tables D–G in S1 Appendix. The number of cancers diagnosed and deaths from lung cancer are presented by follow-up period in Table H in S1 Appendix.

We found that age, smoking duration (years), and pack-years of smoking, drove most predictions. This led us to focus our analyses on developing 2 models: UCL-D and UCL-I, that used just these 3 variables. UCL-D predicts the five-year risk of dying from lung cancer and was a weighted ensemble consisting of 4 modelling algorithms: AdaBoost [48,49], LightGBM [50], Logistic Regression, and Linear Discriminant Analysis. UCL-I predicts the five-year risk of developing lung cancer and included AdaBoost [48,49], LightGBM [50], Bagging, and Cat-Boost [51] algorithms. Details of the ensemble pipelines, their weightings and algorithm hyperparameters are presented Figs H and I and Tables I and J in S1 Appendix.

### UCL models

In internal and external validation, UCL-D and UCL-I showed good discrimination (Tables 2 and 3 and Table K in S1 Appendix), overall performance (Tables L and M in S1 Appendix), and calibration (Fig 2, Table 4, and Table N in S1 Appendix), both overall and across subgroups. In external validation in the PLCO radiography arm, UCL-D had an AUC of 0.803 (95% CI: 0.783, 0.824), an expected/observed (E/O) ratio of 1.05 (95% CI: 0.95, 1.19), and a Brier score of 0.0084 (95% CI: 0.0075, 0.0093). UCL-I had an AUC of 0.787 (95% CI: 0.771, 0.802), an E/O ratio of 1.0 (0.92, 1.07), and a Brier score of 0.0153 (0.0142, 0.0164).

### Discrimination

Despite using approximately one-quarter of the variables, UCL-D achieved parity in discrimination with the LCDRAT (AUC: 0.811, 95%: 0.793, 0.829, $p = 0.18$ for difference with UCL-D). UCL-I achieved parity with PLCOm2012 (AUC: 0.792, 95% CI: 0.779, 0.808, $p = 0.15$ for difference in AUCs) and showed greater discrimination than LLP versions 2 and 3 ($p < 0.001$).

**Table 1. Descriptive characteristics of the development and validation cohorts.**

| | Development cohorts | | Validation cohort |
|---|---|---|---|
| Characteristic | UK Biobank *n* = 216,714 | NLST controls *n* = 26,616 | PLCO chest radiography arm *n* = 40,593 |
| Age [*n*, (%)] | | | |
| <50 | 43,170 (19.92) | - | - |
| 50–54 | 30,077 (13.88) | - | - |
| 55–59 | 39,539 (18.24) | 11,384 (42.77) | 13,965 (34.41) |
| 60–64 | 57,295 (26.44) | 8,170 (30.7) | 12,623 (31.1) |
| 65–69 | 45,520 (21.0) | 4,741 (17.81) | 9,117 (22.46) |
| ≥70 | 1,113 (0.51) | 2,321 (8.72) | 4,879 (12.02) |
| *Missing* | 0 (0.0) | 0 (0.0) | 9 (0.02) |
| Sex–Female [*n*, (%)] | 103,698 (47.85) | 10,919 (41.02) | 16,892 (41.61) |
| *Missing* | 0 (0.0) | 0 (0.0) | 0 (0.0) |
| Ethnicity–White [*n*, (%)] | 208,255 (96.47) | 24,165 (91.50) | 35,818 (88.29) |
| *Missing* | 830 (0.38) | 206 (0.77) | 23 (0.06) |
| Highest qualification [*n*, (%)] | | | |
| Degree | 59,705 (28.07) | 8,213 (31.03) | 13,149 (32.44) |
| Some college | 16,501 (7.76) | 6,072 (22.94) | 9,434 (23.27) |
| Post-secondary school | 33,588 (15.79) | 10,100 (38.17) | 14,403 (35.53) |
| Secondary school | 57,646 (27.11) | 1,211 (4.58) | 3,083 (7.61) |
| None of the above | 45,231 (21.27) | 868 (3.28) | 464 (1.14) |
| *Missing* | 4043 (1.87) | 152 (0.57) | 60 (0.15) |
| Household income (GBP £) | | | |
| <18,000 | 49,067 (26.45) | - | - |
| 18,000–30,999 | 49,023 (26.42) | - | - |
| 31,000–51,999 | 46,120 (24.86) | - | - |
| 52,000–100,000 | 33,020 (17.8) | - | - |
| >100,000 | 8,296 (4.47) | - | - |
| *Missing* | 31,188 (14.39) | - | - |
| Body mass index | | | |
| <18.5 | 1,084 (0.50) | 240 (0.91) | 310 (0.77) |
| 18.5–24 | 62,715 (29.1) | 7,302 (27.65) | 12,743 (31.78) |
| 25–29 | 94,272 (43.75) | 11,442 (43.33) | 17,280 (43.1) |
| 30–34 | 41,469 (19.24) | 5,219 (19.76) | 7,035 (17.55) |
| ≥35 | 15,954 (7.40) | 2,205 (8.35) | 2,726 (6.80) |
| *Missing* | 1,220 (0.56) | 208 (0.78) | 499 (1.23) |
| Smoking status | | | |
| Former | 164,714 (76.01) | 13,764 (51.71) | 8,073 (19.89) |
| Current | 52,000 (23.99) | 12,852 (48.29) | 32,520 (80.11) |
| *Missing* | 0 (0.0) | 0 (0.0) | 0 (0.0) |
| Pack-years of smoking [*n*, (%)] | | | |
| <10 | 35,222 (23.59) | 0 (0.0) | 6,609 (16.63) |
| 11–19 | 39,914 (26.73) | 0 (0.0) | 7,605 (19.13) |
| 20–29 | 29,471 (19.74) | 4 (0.02) | 5,839 (14.69) |
| 30–39 | 20,596 (13.79) | 6,865 (25.79) | 5,108 (12.85) |
| ≥40 | 24,125 (16.16) | 19,747 (74.19) | 14,592 (36.71) |
| *Missing* | 67,386 (31.09) | 0 (0.0) | 840 (2.07) |
| Personal history of cancer [*n*, (%)] | 19,386 (8.95) | 1,197 (4.5) | 1,837 (4.53) |

*(Continued)*

**Table 1.** (Continued)

|  | Development cohorts | | Validation cohort |
| --- | --- | --- | --- |
| *Missing* | 0 (0.0) | 0 (0.0) | 5 (0.01) |
| COPD/Emphysema/Bronchitis [*n*, (%)] | 6,616 (3.06) | 4,617 (17.35) | 3,617 (8.91) |
| *Missing* | 454 (0.21) | 0 (0.0) | 0 (0.0) |
| Family history of lung cancer [*n*, (%)] | 28,765 (13.52) | 5,734 (21.54) | 4,566 (11.71) |
| *Missing* | 3,944 (1.82) | 0 (0.0) | 1602 (3.95) |

GBP, British Pounds; COPD, Chronic Obstructive Pulmonary Disease; NLST, National Lung Screening Trial; PLCO, Prostate, Lung, Colorectal, and Ovarian Cancer Screening Trial.

## Calibration

The UCL models were well calibrated across risk thresholds at which eligibility for screening is typically set, tending modestly towards underprediction in the highest risk decile in the PLCO radiography arm (Fig 2). By contrast, PLCOm2012 and LCRAT tended modestly towards underprediction at deciles corresponding to observed risks of 1% to 4%, which is more clinically disadvantageous than overprediction. As the PLCOm2012, LCDRAT, and LCRAT models were developed in the control arm of the PLCO trial, the strong relative performance of the UCL models is notable. All models modestly overpredicted risk in the UK Biobank cohort, with the extent of overprediction most notable for the LLP version 2.

## Overall performance

When considering Brier scores, an overall measure of model performance comparing the closeness of predicted probabilities and observed outcomes [37], there was little or no distinction between the models in the UK Biobank and PLCO radiography arm (Tables L and M in S1 Appendix). In the PLCO radiography arm, both models predicting the five-year risk of death, UCL-D and LCDRAT had a Brier score of 0.0084 (95% CI: 0.0075, 0.0093). Brier scores vary with prevalence; consequently, models predicting the risk of developing lung cancer had higher scores. Nevertheless, the same pattern was observed: UCL-I had a Brier score of 0.0153 (95% CI: 0.0142, 0.0164), LCRAT a score of 0.0152 (95% CI: 0.0143, 0.0164), and LLP version 2 a score of 0.0153 (95% CI: 0.0143, 0.0165).

## Risk thresholds to select individuals for screening

Using the USPSTF-2021 criteria, 34,654 (43.0%) of the entire PLCO dataset would be eligible for lung cancer screening. All UCL models had higher sensitivity than the USPSTF-2021 at an equivalent specificity, with the gains in sensitivity higher when predicting five-year risk of death from lung cancer (Table O in S1 Appendix). For UCL-I at a five-year risk threshold of 1.17%, the gains in sensitivity were 6.2% relative to the USPSTF-2021 criteria (83.9% [95% CI: 82.0, 86.1%] versus 77.7% [95% CI: 75.8, 80.2%]). By contrast, UCL-D at a five-year risk threshold of 0.68% would lead to a 7.9% increase in sensitivity (85.5% [95% CI: 82.8, 88.2%] versus 77.5% [95% CI: 74.6, 80.9%]) for the same specificity.

At the aforementioned risk cut-offs, 96.2% of individuals selected by UCL-D would also have been eligible for screening with UCL-I. By 10-years of follow-up, those selected for screening with UCL-D but not UCL-I tended towards a greater risk of developing and dying from lung cancer than those selected by UCL-I but not UCL-D, though this trend was not statistically significant (Fig J in S1 Appendix; Logrank test: *p* = 0.15 for differences in lung cancer deaths and *p* = 0.41 for differences in lung cancers).

**Table 2. Discriminative accuracy (AUC) overall and by subgroup in the PLCO chest radiography cohort.**

| | AUC with 95% confidence intervals | | | | | |
|---|---|---|---|---|---|---|
| | **UCL-D** | **LCDRAT** | **UCL-I** | **LCRAT** | **PLCOm2012** | **LLPv2** |
| Overall | 0.803 (0.783, 0.824) | 0.811 (0.793, 0.829) | 0.787 (0.771, 0.802) | 0.798 (0.784, 0.814) | 0.792 (0.779, 0.808) | 0.743 (0.726, 0.762) |
| Age category | | | | | | |
| 55–59 | 0.800 (0.745, 0.844) | 0.815 (0.766, 0.858) | 0.797 (0.762, 0.833) | 0.817 (0.778, 0.847) | 0.794 (0.756, 0.825) | 0.729 (0.695, 0.767) |
| 60–64 | 0.793 (0.753, 0.831) | 0.799 (0.764, 0.830) | 0.759 (0.722, 0.790) | 0.776 (0.742, 0.804) | 0.770 (0.741, 0.796) | 0.716 (0.678, 0.751) |
| 65–69 | 0.787 (0.747, 0.823) | 0.806 (0.768, 0.840) | 0.781 (0.752, 0.809) | 0.792 (0.765, 0.823) | 0.798 (0.775, 0.823) | 0.747 (0.715, 0.777) |
| 70–74 | 0.747 (0.694, 0.790) | 0.725 (0.673, 0.773) | 0.728 (0.685, 0.768) | 0.723 (0.677, 0.760) | 0.720 (0.682, 0.753) | 0.675 (0.628, 0.715) |
| Sex | | | | | | |
| Female | 0.800 (0.771, 0.828) | 0.801 (0.771, 0.831) | 0.771 (0.745, 0.796) | 0.784 (0.760, 0.804) | 0.784 (0.764, 0.805) | 0.731 (0.699, 0.757) |
| Male | 0.803 (0.773, 0.828) | 0.818 (0.791, 0.841) | 0.795 (0.774, 0.814) | 0.807 (0.789, 0.825) | 0.798 (0.776, 0.819) | 0.755 (0.731, 0.779) |
| Smoking status | | | | | | |
| Former | 0.813 (0.787, 0.842) | 0.819 (0.793, 0.843) | 0.791 (0.768, 0.814) | 0.802 (0.781, 0.824) | 0.793 (0.774, 0.814) | 0.741 (0.715, 0.768) |
| Current | 0.681 (0.642, 0.721) | 0.705 (0.667, 0.744) | 0.677 (0.650, 0.717) | 0.698 (0.672, 0.736) | 0.694 (0.669, 0.724) | 0.651 (0.622, 0.696) |
| Qualifications | | | | | | |
| Degree | 0.680 (0.323, 0.921) | 0.709 (0.427, 0.898) | 0.610 (0.455, 0.779) | 0.681 (0.551, 0.799) | 0.629 (0.509, 0.751) | 0.609 (0.493, 0.742) |
| Some college | 0.756 (0.654, 0.834) | 0.796 (0.688, 0.900) | 0.750 (0.686, 0.818) | 0.771 (0.698, 0.848) | 0.726 (0.640, 0.803) | 0.663 (0.597, 0.737) |
| Post-secondary | 0.730 (0.632, 0.825) | 0.772 (0.651, 0.859) | 0.753 (0.688, 0.826) | 0.780 (0.718, 0.843) | 0.763 (0.704, 0.827) | 0.741 (0.670, 0.814) |
| Secondary school | 0.638 (0.542, 0.719) | 0.650 (0.550, 0.742) | 0.620 (0.545, 0.691) | 0.644 (0.578, 0.710) | 0.664 (0.608, 0.718) | 0.643 (0.582, 0.707) |
| None of above | 0.700 (0.671, 0.725) | 0.699 (0.675, 0.728) | 0.673 (0.651, 0.697) | 0.689 (0.667, 0.710) | 0.693 (0.670, 0.714) | 0.644 (0.621, 0.667) |
| Ethnicity | | | | | | |
| Asian | 0.839 (0.805, 0.874) | 0.857 (0.825, 0.888) | 0.804 (0.770, 0.834) | 0.823 (0.791, 0.848) | 0.820 (0.787, 0.848) | 0.752 (0.714, 0.784) |
| Black | 0.805 (0.754, 0.847) | 0.802 (0.751, 0.842) | 0.804 (0.771, 0.835) | 0.811 (0.778, 0.841) | 0.800 (0.764, 0.828) | 0.748 (0.709, 0.790) |
| Other | 0.791 (0.754, 0.822) | 0.789 (0.757, 0.818) | 0.765 (0.737, 0.793) | 0.774 (0.749, 0.802) | 0.765 (0.741, 0.791) | 0.728 (0.700, 0.755) |
| White | 0.734 (0.671, 0.795) | 0.744 (0.675, 0.805) | 0.735 (0.696, 0.779) | 0.741 (0.697, 0.785) | 0.755 (0.715, 0.791) | 0.707 (0.658, 0.758) |
| Body mass index | | | | | | |
| <18.5 | 0.622 (0.477, 0.747) | 0.574 (0.429, 0.75) | 0.742 (0.667, 0.822) | 0.716 (0.578, 0.832) | 0.705 (0.569, 0.820) | 0.775 (0.651, 0.862) |
| 18.5–24 | 0.802 (0.766, 0.832) | 0.804 (0.769, 0.833) | 0.791 (0.763, 0.816) | 0.792 (0.764, 0.818) | 0.786 (0.757, 0.807) | 0.753 (0.724, 0.781) |
| 25–29 | 0.804 (0.767, 0.834) | 0.807 (0.775, 0.835) | 0.774 (0.749, 0.798) | 0.792 (0.765, 0.815) | 0.792 (0.768, 0.811) | 0.731 (0.705, 0.755) |
| 30–34 | 0.817 (0.772, 0.863) | 0.840 (0.804, 0.872) | 0.792 (0.745, 0.837) | 0.817 (0.775, 0.853) | 0.795 (0.754, 0.835) | 0.742 (0.689, 0.799) |
| ≥35 | 0.780 (0.631, 0.902) | 0.748 (0.539, 0.904) | 0.801 (0.721, 0.872) | 0.792 (0.686, 0.877) | 0.811 (0.725, 0.884) | 0.738 (0.647, 0.819) |
| COPD | | | | | | |
| No | 0.808 (0.785, 0.829) | 0.812 (0.792, 0.832) | 0.786 (0.768, 0.804) | 0.796 (0.779, 0.813) | 0.790 (0.775, 0.806) | 0.749 (0.731, 0.767) |
| Yes | 0.698 (0.641, 0.750) | 0.694 (0.638, 0.745) | 0.708 (0.666, 0.747) | 0.710 (0.664, 0.746) | 0.717 (0.681, 0.753) | 0.671 (0.632, 0.723) |
| Previous cancer | | | | | | |
| No | 0.805 (0.784, 0.826) | 0.814 (0.795, 0.832) | 0.790 (0.774, 0.807) | 0.801 (0.785, 0.817) | 0.794 (0.781, 0.811) | 0.749 (0.732, 0.769) |
| Yes | 0.762 (0.679, 0.841) | 0.751 (0.651, 0.816) | 0.718 (0.655, 0.790) | 0.738 (0.674, 0.798) | 0.741 (0.682, 0.799) | 0.687 (0.608, 0.765) |
| Family history of lung cancer | | | | | | |
| No | 0.798 (0.773, 0.820) | 0.807 (0.786, 0.826) | 0.786 (0.768, 0.804) | 0.796 (0.780, 0.812) | 0.793 (0.779, 0.810) | 0.746 (0.727, 0.765) |
| Yes | 0.826 (0.776, 0.864) | 0.822 (0.770, 0.860) | 0.780 (0.739, 0.815) | 0.787 (0.740, 0.820) | 0.770 (0.729, 0.803) | 0.715 (0.670, 0.760) |

The discrimination of the UCL models was consistent across most subgroups. Notably, all models had lower AUCs in current relative to former smokers among those with a BMI <18.5 and those with COPD. Discrimination was good across all ethnic groups in the UCL models.

AUC, area under the receiver operating curve; COPD, chronic obstructive pulmonary disease; PLCO, Prostate, Lung, Colorectal and Ovarian Cancer Screening Trial; LLPv2, Liverpool Lung Project version 2; LCDRAT, Lung Cancer Death Risk Assessment Tool; LCRAT, Lung Cancer Risk Assessment Tool; UCL-D predicts lung cancer death; UCL-I predicts occurrence of lung cancer.

**Table 3. Discrimination of UCL-D, Cox models, and the constrained LCDRAT, LCRAT, and PLCOm2012 models.**

| | Predictors | UK Biobank | PLCO chest radiography arm |
|---|---|---|---|
| | n | AUC (95% CI) | AUC (95% CI) |
| UCL-D | 3 | 0.826 (0.815, 0.838) | 0.803 (0.783, 0.824) |
| Cox model (no interactions) | 3 | 0.817 (0.809, 0.825) | 0.782 (0.772, 0.793) |
| Cox model with interactions | 3 | 0.819 (0.811, 0.827) | 0.785 (0.775, 0.795) |
| LCDRAT-constrained | 6 | 0.821 (0.807, 0.833) | 0.801 (0.782, 0.820) |
| LCRAT-constrained | 6 | 0.806 (0.796, 0.819) | 0.788 (0.773, 0.803) |
| PLCOm2012-constrained | 5 | 0.786 (0.772, 0.799) | 0.778 (0.766, 0.797) |

AUC, area under the receiver operating curve; CI, confidence intervals; PLCO, Prostate, Lung, Colorectal, and Ovarian Cancer Screening Trial; LCDRAT, Lung Cancer Death Risk Assessment Tool; LCRAT, Lung Cancer Risk Assessment Tool. UCL-D, the 2 Cox models, and LCDRAT-constrained predict 5-year risk of lung cancer death; LCRAT-constrained and PLCOm2012 constrained 5-year risk of lung cancer occurrence. Cox models were modelled with restricted cubic splines, with and without mutual interactions between age, smoking duration, and pack-years using the same development dataset as UCL-D. The LCDRAT and LCRAT-constrained models use age, sex, quit-years, smoking duration, cigarettes per day, and pack-years. PLCOm2012-constrained uses age, smoking status, smoking duration, cigarettes per day, and quit-years. Both the LCRAT/LCDRAT and PLCOm2012 models were developed in the control arm of the PLCO trial. The relatively shallow drop-off in discriminatory performance between the various constrained models and their full versions show the relative importance of few smoking parameters and validates our findings that few smoking variables drive all lung cancer models in ever-smokers. The improvement seen by UCL-D over Cox models using the same data and variables reflects the statistical advantages of ensemble machine learning approaches.

Using decision curve analysis, at all risk thresholds, the net benefit of the UCL models is greater than screening using the USPSTF-2021 criteria (Fig 3 and Fig K in S1 Appendix). At suggested risk thresholds, the net benefit of compared risk models other than LLP are equivalent.

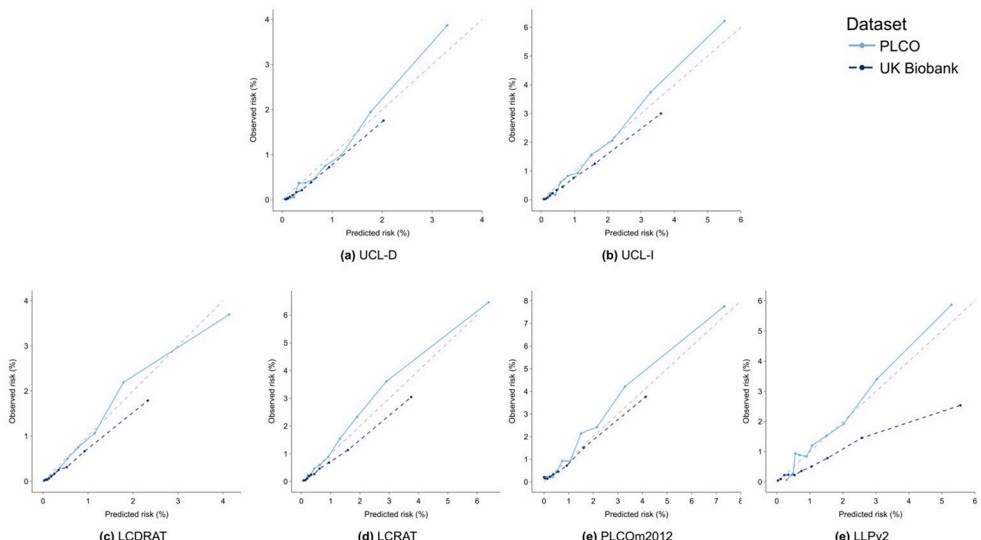

**Fig 2. Calibration curves.** Calibration curves showing UCL and comparator models in the UK Biobank (dark blue dashed lines) and US PLCO Cancer Screening Trial chest radiography arm (light blue line). The 45-degree lines in grey indicate perfect calibration. Curves were generated by splitting individuals into 10 risk deciles based on their predicted risk. Each curve shows the mean predicted risk against the observed risk by risk decile. Observed risk was calculated using a Kaplan–Meier estimator. The UCL models showed good calibration in external validation in the PLCO intervention arm, particularly at predicted risk between 1% and 2% at which risk thresholds are commonly set. At these thresholds, there was modest underprediction with the LCDRAT, LCRAT, and PLCOm2012 models in the PLCO intervention arm. All models modestly overpredicted risk in the UK Biobank, with the exception of the Liverpool Lung Project (LLPv2) version 2 model, which strongly overpredicted risk. LCDRAT, Lung Cancer Death Risk Assessment Tool; LCRAT, Lung Cancer Risk Assessment Tool. UCL-D predicts lung cancer death; UCL-I predicts occurrence of lung cancer.

**Table 4. Calibration overall and by subgroup in the PLCO chest radiography cohort.**

| | Ratio of expected-to-observed cancers with 95% confidence intervals | | | | | |
|---|---|---|---|---|---|---|
| | **UCL-D** | **LCDRAT** | **UCL-I** | **LCRAT** | **PLCOm2012** | **LLPv2** |
| Overall | 1.05 (0.95, 1.19) | 1.09 (0.99, 1.24) | 1.00 (0.92, 1.07) | 0.96 (0.89, 1.03) | 0.93 (0.86, 0.99) | 0.97 (0.90, 1.04) |
| Age category | | | | | | |
| 55–59 | 1.15 (0.93, 1.45) | 1.08 (0.88, 1.36) | 1.05 (0.91, 1.23) | 1.01 (0.86, 1.18) | 0.91 (0.78, 1.05) | 0.76 (0.66, 0.89) |
| 60–64 | 1.15 (0.97, 1.35) | 1.21 (1.02, 1.43) | 1.12 (0.97, 1.31) | 1.08 (0.93, 1.25) | 0.98 (0.87, 1.14) | 1.06 (0.93, 1.25) |
| 65–69 | 0.96 (0.8, 1.18) | 1.02 (0.85, 1.26) | 0.86 (0.75, 0.99) | 0.85 (0.75, 0.97) | 0.85 (0.75, 0.96) | 0.92 (0.80, 1.04) |
| 70–74 | 1.00 (0.81, 1.26) | 1.08 (0.89, 1.38) | 1.01 (0.88, 1.22) | 0.95 (0.83, 1.15) | 1.01 (0.87, 1.18) | 1.15 (0.99, 1.39) |
| Sex | | | | | | |
| Female | 1.13 (0.95, 1.38) | 1.38 (1.16, 1.68) | 1.00 (0.88, 1.12) | 1.07 (0.93, 1.19) | 0.99 (0.88, 1.09) | 0.96 (0.84, 1.07) |
| Male | 1.01 (0.87, 1.16) | 0.94 (0.81, 1.07) | 1.00 (0.90, 1.09) | 0.90 (0.81, 0.99) | 0.90 (0.81, 0.98) | 0.99 (0.89, 1.07) |
| Smoking status | | | | | | |
| Former | 1.14 (1.01, 1.34) | 1.02 (0.90, 1.19) | 1.09 (1.00, 1.21) | 0.89 (0.81, 0.99) | 0.92 (0.85, 1.00) | 1.16 (1.06, 1.29) |
| Current | 0.93 (0.81, 1.10) | 1.18 (1.04, 1.40) | 0.88 (0.79, 0.97) | 1.06 (0.94, 1.16) | 0.95 (0.85, 1.03) | 0.73 (0.64, 0.80) |
| Qualifications | | | | | | |
| Degree | 1.19 (0.99, 1.52) | 1.00 (0.82, 1.26) | 1.14 (0.99, 1.37) | 0.93 (0.80, 1.12) | 0.90 (0.79, 1.07) | 1.18 (1.02, 1.43) |
| Some college | 1.17 (0.94, 1.48) | 1.14 (0.91, 1.45) | 1.07 (0.92, 1.25) | 0.98 (0.85, 1.15) | 0.94 (0.82, 1.10) | 1.00 (0.87, 1.17) |
| Post-secondary | 1.03 (0.89, 1.22) | 1.18 (1.02, 1.40) | 0.98 (0.86, 1.11) | 1.01 (0.89, 1.14) | 0.99 (0.88, 1.10) | 0.94 (0.82, 1.07) |
| Secondary school | 0.80 (0.64, 1.06) | 1.04 (0.83, 1.39) | 0.77 (0.65, 0.95) | 0.88 (0.75, 1.11) | 0.86 (0.73, 1.04) | 0.75 (0.62, 0.92) |
| None of above | 0.60 (0.34, 1.28) | 0.77 (0.44, 1.69) | 0.69 (0.44, 1.50) | 0.79 (0.50, 1.68) | 0.82 (0.52, 1.62) | 0.71 (0.44, 1.52) |
| Ethnicity | | | | | | |
| Asian | 1.08 (0.66, 2.52) | 0.72 (0.45, 1.69) | 1.17 (0.76, 2.41) | 0.73 (0.48, 1.53) | 0.69 (0.44, 1.18) | 1.27 (0.83, 2.66) |
| Black | 0.59 (0.45, 0.86) | 0.86 (0.66, 1.26) | 0.61 (0.49, 0.78) | 0.73 (0.59, 0.93) | 0.78 (0.62, 0.96) | 0.66 (0.53, 0.86) |
| Other | 0.73 (0.48, 1.39) | 0.47 (0.31, 0.90) | 0.92 (0.63, 1.53) | 0.56 (0.39, 0.93) | 0.70 (0.49, 1.15) | 0.93 (0.65, 1.53) |
| White | 1.12 (1.00, 1.27) | 1.16 (1.04, 1.31) | 1.04 (0.96, 1.12) | 1.01 (0.93, 1.09) | 0.96 (0.89, 1.04) | 1.00 (0.93, 1.09) |
| Body mass index | | | | | | |
| <18.5 | 0.54 (0.32, 1.27) | 1.21 (0.74, 2.81) | 0.70 (0.47, 1.44) | 1.09 (0.74, 2.21) | 1.13 (0.76, 2.25) | 0.63 (0.42, 1.24) |
| 18.5–24 | 0.92 (0.80, 1.14) | 1.13 (0.98, 1.39) | 0.89 (0.79, 1.02) | 0.99 (0.89, 1.15) | 0.92 (0.84, 1.07) | 0.88 (0.79, 1.02) |
| 25–29 | 1.12 (0.96, 1.35) | 1.09 (0.94, 1.30) | 1.00 (0.89, 1.12) | 0.91 (0.81, 1.01) | 0.91 (0.82, 1.02) | 0.98 (0.87, 1.11) |
| 30–34 | 1.08 (0.85, 1.47) | 0.95 (0.74, 1.27) | 1.17 (1.00, 1.43) | 0.97 (0.83, 1.19) | 0.93 (0.80, 1.10) | 1.11 (0.94, 1.35) |
| ≥35 | 1.81 (1.11, 4.12) | 1.36 (0.84, 3.11) | 1.56 (1.03, 2.73) | 1.12 (0.74, 1.96) | 1.19 (0.78, 2.05) | 1.43 (0.95, 2.51) |
| COPD | | | | | | |
| No | 1.14 (1.04, 1.32) | 1.07 (0.96, 1.23) | 1.08 (0.99, 1.17) | 0.93 (0.86, 1.01) | 0.93 (0.86, 1.00) | 1.08 (0.99, 1.17) |
| Yes | 0.71 (0.58, 0.90) | 1.18 (0.97, 1.49) | 0.69 (0.58, 0.84) | 1.07 (0.89, 1.30) | 0.96 (0.81, 1.13) | 0.59 (0.49, 0.72) |
| Previous cancer | | | | | | |
| No | 1.06 (0.95, 1.20) | 1.10 (0.99, 1.24) | 1.00 (0.92, 1.09) | 0.96 (0.88, 1.04) | 0.91 (0.84, 0.98) | 0.94 (0.86, 1.02) |
| Yes | 0.91 (0.65, 1.46) | 1.03 (0.73, 1.66) | 0.88 (0.68, 1.21) | 0.89 (0.69, 1.23) | 1.26 (0.96, 1.73) | 1.51 (1.17, 2.06) |
| Family history of lung cancer | | | | | | |
| No | 1.12 (1.00, 1.29) | 1.08 (0.97, 1.25) | 1.07 (0.99, 1.18) | 0.96 (0.89, 1.06) | 0.91 (0.84, 0.99) | 1.00 (0.93, 1.10) |
| Yes | 0.73 (0.57, 0.95) | 1.16 (0.91, 1.50) | 0.65 (0.55, 0.81) | 0.94 (0.80, 1.16) | 1.01 (0.87, 1.28) | 0.83 (0.70, 1.04) |

Ideal calibration is 1, where the expected and observed number of cancers is the same. The UCL models were well calibrated across most subgroups. Specifically, calibration was good across age groups, sexes, and by smoking status. As the number of outcomes in sub-groups was very small, in certain sub-groups the confidence intervals were wide. Notably, all models underpredicted risk in at least 1 ethnic subgroup. Of predictors used in other models, but not in the UCL models, there was some underprediction of risk among those with COPD and a family history of lung cancer.

COPD, chronic obstructive pulmonary disease; PLCO, Prostate, Lung, Colorectal and Ovarian Cancer Screening Trial; LLPv2, Liverpool Lung Project model version 2; LCDRAT, Lung Cancer Death Risk Assessment Tool; LCRAT, Lung Cancer Risk Assessment Tool. UCL-D predicts lung cancer death; UCL-I predicts occurrence of lung cancer.

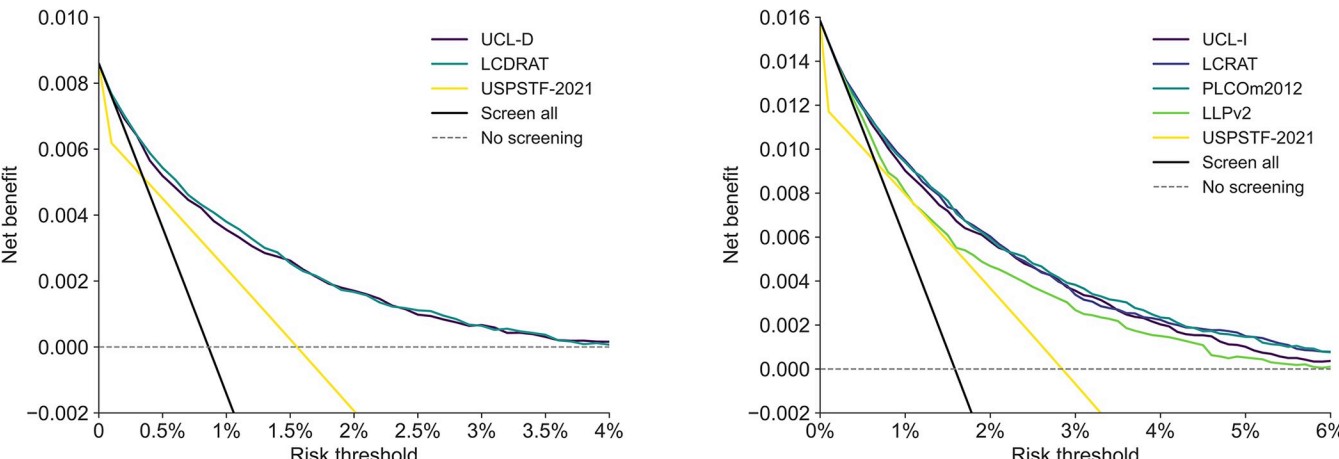

**Fig 3. Decision curves of selected models in the PLCO validation cohort.** Net benefit across a range of thresholds of models predicting five-year risk of death from lung cancer (A) and developing lung cancer (B) compared against US Preventive Services Taskforce (USPSTF) 2021 screening eligibility criteria in the PLCO Cancer Screening Trial chest radiography arm validation dataset. The PLCOm2012 model predicts six-year risk of lung cancer. As the performance of PLCOm2012 over a five-year timeframe was similar to that of six-years, for comparability, predictions over a five-year timeframe are shown here. All models studied except the Liverpool Lung Project (LLPv2) version 2 had a greater net clinical benefit than using the USPSTF-2021 criteria for screening eligibility across all risk thresholds. All other risk models had a comparable net benefit to each other. LCDRAT, Lung Cancer Death Risk Assessment Tool; LCRAT, Lung Cancer Risk Assessment Tool. UCL-D predicts lung cancer death; UCL-I predicts occurrence of lung cancer.

## Discussion

We have developed parsimonious models for lung cancer screening that combine the simplicity of existing risk factor-based criteria with the predictive performance of complex risk prediction models. Furthermore, we show in benchmarking comparisons that ensemble machine learning models with 3 predictors—age, smoking duration, and smoking pack-years—have equivalent predictive performance and clinical usefulness to existing models requiring 11 predictors.

In this analysis, we used ensemble machine learning to leverage the predictions of several optimised model pipelines. Ensemble modelling is based on the concept that different models make different types of mistake, and their errors begin to cancel each other out, such that combining these statistical models could be expected to improve the performance that any one might achieve [52]. By iteratively trialling and optimising a wide range of modelling approaches before subsequently creating ensembles of these approaches, AutoPrognosis ensures that the strongest performing model for that dataset will be derived and allows reproducibility by transparently showing how models were selected. This avoids the need to develop multiple independent models.

In the UK, eligibility for National Health Service screening pilots is based on meeting either a five-year absolute risk of lung cancer of ≥2.5% with the LLP risk score or a six-year absolute risk of ≥1.51% with the PLCOm2012 [23]. The use of 2 risk scores where eligibility differs by more than a percentage point in predicted absolute risk, and where a higher risk is tolerated over a five-year period than a six-year period, highlights the policy challenge in adopting the optimal risk-based approach for a particular setting. This approach requires the collection of 17 different unique predictors, as well as the mapping of US educational levels and US ethnicity categorisations to the UK. With an estimated 7 million current smokers in the UK [25]—even ignoring former smokers—the time and resource requirements to determine screening eligibility at a population scale will be challenging. Using 3 unambiguous variables but with equivalent or improved performance, the UCL models could be completed more easily online or in primary healthcare, simplifying the implementation of lung cancer screening.

A potential risk with using only few predictors is that the models will underperform in different subgroups. Across all major subgroups the performance of the UCL models was equivalent to existing models and importantly this included all 4 ethnic groups available in this analysis. The UCL models were also well calibrated in different subgroups, with no sex, smoking status, or age differences. However, there was some undercalibration in 2 groups used as predictors in comparator models: a history of COPD and family history of lung cancer. Furthermore, all models were undercalibrated in at least 1 ethnic group. As discrimination was good for most models, this suggests that lower thresholds could be considered, particularly in black populations, analogous to the relaxation in the age and smoking intensity criteria made between the USPSTF-2013 and USPSTF-2021 screening recommendations [53]. More work is required to improve calibration among different ethnicities, although it is notable that simply including this predictor in models had limited impact.

In keeping with Katki and colleagues [19], we found that UCL-D, predicting the risk of death from lung cancer, had greater discrimination than models predicting lung cancer occurrence. In these analyses, there was >96% overlap between UCL-D and UCL-I in terms of those selected for screening, with those selected by UCL-D but not UCL-I showing a trend towards a greater risk of death from lung cancer with longer follow-up (Fig J in S1 Appendix). In microsimulation modelling, overall outcomes differed little between a model predicting death from lung cancer compared with models predicting developing lung cancer [13]. Given this, UCL-D would be the more appropriate model to consider for implementation.

This study has strengths. We used large prospective cohorts for model development and validation. Our external validation cohort is both temporally and geographically distinct. We used robust methods for model development and internal validation while externally validating our models extensively using multiple approaches and in a wide range of subgroups. Further, we benchmarked our models against leading comparators. Moreover, by using few, unambiguous, variables, our models could be widely applied after further validation and, where necessary, recalibration. Finally, we have made our models openly available for independent assessment.

This study has several limitations. We have used retrospective data, such that findings may differ if used to prospectively determine screening eligibility. However, both the PLCOm2012 and the LLP models have been studied in prospective settings, establishing the benefits of risk-model against risk-factor-based screening. By benchmarking against these models, we can be confident in the performance of our models in a screening programme. To confirm the generalisability of our models, validation in datasets from beyond the US and UK will be the subject of further work. Our analyses have been performed in research cohorts rather than routinely collected electronic health records that may better reflect the broader population. In keeping with the models used as benchmark comparators in this work, the UCL models may not perform to the same extent in routinely collected electronic health records as smoking data are not usually present in the same depth [54]. Nevertheless, screening programmes are unlikely to rely on existing electronic health records given known challenges of missing and inaccurately coded predictors [55]. Finally, our risk models exclude never-smokers. To date, no risk model has been able to discriminate those never smokers with sufficient risk to meet existing criteria for lung cancer screening.

In conclusion, we used ensemble machine learning to explicitly maximise model parsimony, an approach that holds promise in multiple disease areas. Our prediction models to determine lung cancer screening eligibility require only 3 variables—age, smoking duration, and pack-years—and perform at or above parity with existing risk models in use. Further validation in alternative datasets as well as prospective implementation should be considered.

## Supporting information

**S1 Appendix. Supporting Information.**
(PDF)

## Acknowledgments

This research has been conducted using the UK Biobank Resource under application number 68073 and we wish to thank all participants in the included studies, as well as the National Cancer Institute for access to NCI's data collected by the National Lung Screening Trial (NLST) and Prostate, Lung, Colorectal and Ovarian Cancer Screening Trial (PLCO). We also wish to thank Arjun Nair and Sujin Kang for their feedback on earlier versions of this project, as well as Stephen Duffy for his comments on this work.

The statements contained herein are solely those of the authors and do not represent or imply concurrence or endorsement by NCI.

## Author Contributions

**Conceptualization:** Thomas Callender, Sam M. Janes.

**Data curation:** Thomas Callender.

**Formal analysis:** Thomas Callender.

**Funding acquisition:** Thomas Callender.

**Investigation:** Thomas Callender.

**Methodology:** Thomas Callender, Fergus Imrie, Nora Pashayan, Mihaela van der Schaar.

**Project administration:** Thomas Callender.

**Software:** Thomas Callender, Bogdan Cebere, Mihaela van der Schaar.

**Supervision:** Nora Pashayan, Mihaela van der Schaar, Sam M. Janes.

**Validation:** Thomas Callender.

**Writing – original draft:** Thomas Callender.

**Writing – review & editing:** Thomas Callender, Fergus Imrie, Nora Pashayan, Neal Navani, Mihaela van der Schaar, Sam M. Janes.

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
