## [Editor Report · Decision Letter 0]

21 Mar 2023

Dear Dr Callender, 

Thank you for submitting your manuscript entitled "Assessing eligibility for lung cancer screening:

Parsimonious multi-country ensemble machine learning models for lung cancer prediction" for consideration by PLOS Medicine.

Your manuscript has now been evaluated by the PLOS Medicine editorial staff as well as by an academic editor with relevant expertise and I am writing to let you know that we would like to send your submission out for external peer review.

Please re-submit your manuscript within two working days, i.e. by Mar 23 2023 11:59PM.

Kind regards,

Philippa Dodd, MBBS MRCP PhD

PLOS Medicine

---

## [Decision Letter · Decision Letter 1]

19 May 2023

Dear Dr. Callender,

Thank you very much for submitting your manuscript "Assessing eligibility for lung cancer screening:

Parsimonious multi-country ensemble machine learning models for lung cancer prediction" (PMEDICINE-D-23-00694R1) for consideration at PLOS Medicine. 

Your paper was evaluated by a senior editor and discussed among all the editors here. It was also sent to independent reviewers, including a statistical reviewer. The reviews are appended at the bottom of this email and any accompanying reviewer attachments can be seen via the link below:

[LINK]

In light of these reviews, I am afraid that we will not be able to accept the manuscript for publication in the journal in its current form, but we would like to consider a revised version that addresses the reviewers' and editors' comments. Obviously we cannot make any decision about publication until we have seen the revised manuscript and your response, and we plan to seek re-review by one or more of the reviewers. 

We expect to receive your revised manuscript by Jun 09 2023 11:59PM. Please email us (plosmedicine@plos.org) if you have any questions or concerns.

We look forward to receiving your revised manuscript. 

Sincerely,

Philippa Dodd, MBBS MRCP PhD

PLOS Medicine

plosmedicine.org

GENERAL

Please respond to all editor and reviewer comments detailed below, in full.

Please include line numbers starting at 1 on page 2 and in continuous sequence thereafter.

* The code and model were inaccessible (by some) via the link, please see reviewer comments below and please amend *

ABSTRACT

Please structure your abstract using the PLOS Medicine headings (Background, Methods and Findings, Conclusions).

Please combine the Methods and Findings sections into one section, “Methods and findings”.

Abstract Background: Provide the context of why the study is important.

Abstract Methods and Findings:

Please ensure that all numbers presented in the abstract are present and identical to numbers presented in the main manuscript text.

Please provide further details of the cohorts used in your study including aggregate participant demographics, years over which data were collected and so on.

Please provide (brief) details of the trials that you refer to.

Were your analyses adjusted for any variables? If so, please provide details.

Please quantify the main results with 95% CIs and p values. When reporting p values please report as p<0.001 and where higher as p=0.002, for example. If not, please for the purpose of transparent data reporting please clearly describe the reasons why not. When reporting CIs please use commas instead of hyphens to separate upper and lower bounds as these can be confused with reporting of negative values.

In the last sentence of the Abstract Methods and Findings section, please describe the main limitation(s) of the study's methodology.

AUTHOR SUMMARY

At this stage, we ask that you include a short, non-technical Author Summary of your research to make findings accessible to a wide audience that includes both scientists and non-scientists. The authors summary should consist of 2-3 succinct bullet points under each of the following headings:

• Why Was This Study Done? Authors should reflect on what was known about the topic before the research was published and why the research was needed.

• What Did the Researchers Do and Find? Authors should briefly describe the study design that was used and the study’s major findings. Do include the headline numbers from the study, such as the sample size and key findings. 

• What Do These Findings Mean? Authors should reflect on the new knowledge generated by the research and the implications for practice, research, policy, or public health. Authors should also consider how the interpretation of the study’s findings may be affected by the study limitations. In the final bullet point of ‘What Do These Findings Mean?’, please describe the main limitations of the study in non-technical language.

Author Summary should immediately follow the Abstract in your revised manuscript. This text is subject to editorial change and should be distinct from the scientific abstract. Please see our author guidelines for more information: https://journals.plos.org/plosmedicine/s/revising-your-manuscript#loc-author-summary.

INTRODUCTION

Paragraph 1 sentence beginning ‘In this work…’ suggest removing this to the discussion and tempering the language, ‘state of the art’ for example.

Please address past research and explain the need for and potential importance of your study. Indicate whether your study is novel and how you determined that. If there has been a systematic review of the evidence related to your study (or you have conducted one), please refer to and reference that review and indicate whether it supports the need for your study.

Please conclude the Introduction with a clear description of the study question or hypothesis.

METHODS and RESULTS

Please provide further details of the cohorts and trials from which participant data was extracted, number of years over which data were collected and so on.

In-line with reviewer comments, please see below, please provide justifications/clarifications for your choice/use of the datasets included.

As above (see under abstract), please quantify the main results with 95% CIs and p values. When reporting p values please report as p<0.001 and where higher as p=0.002, for example. When reporting CIs please use commas instead of hyphens to separate upper and lower bounds as these can be confused with reporting of negative values.

Please remove the ‘Clinical Usefulness’ paragraph from the end of the results section and include this in a relevant part of the discussion.

TABLES

Table 1 – ‘Age (n, %)’ – suggest Age [n, (%)] for clarity, please amend throughout the table.

Table 2 – please clearly define the numerical values contained within parentheses for the reader. 

Please include an appropriate caption, including the meaning of the shaded part of the table, which clearly describes the table contents without the need to refer to the text. Please indicate if your analyses are adjusted (and which factors are adjusted for) or unadjusted. Where adjusted analyses are presented, please also present unadjusted analyses for comparison.

FIGURES

Figure 1 – (and throughout) please consider avoiding the use of red and/or green to make your figures more accessible to those with colour blindness. The text in this figure is very small particularly on the axes of the sub-plots (part c), please revise.

Figure 2 – please define the meaning of the grey dashed line for the reader. As above, the text here is very small and rather inaccessible as presented, please revise.

Figure 3 – please define all abbreviations including those used to depict the models.

DISCUSSION

Please present and organize the Discussion as follows: a short, clear summary of the article's findings; what the study adds to existing research and where and why the results may differ from previous research; strengths and limitations of the study; implications and next steps for research, clinical practice, and/or public policy; one-paragraph conclusion. Please avoid the use of sub-headings that the discussion reads as a single piece of continuous prose.

REFERENCES 

For in-text reference callouts citations should be placed in square parentheses preceding punctuation, for example [1,3,6-9]. Please note the absence of spaces between citations.

In the bibliography, please list up to but no more than 6 author names followed by et al.

Please ensure journal name abbreviations should be those found in the National Center for Biotechnology Information (NCBI) databases.

STATEMENTS

Page 25 – please remove the data availability statement and include only in the manuscript submission form, it will be compiled as metadata.

Please move the ethics statement to an appropriate part of the methods section.

Page 26 – please remove the funding statement and include only in the manuscript submission form

SUPPORTING INFORMATION

Please apply all comments detailed above to the supporting information (statistical reporting, referencing format etc) to the supporting files.

eFigure 5& 6 – please indicate in the figure captions the meaning of the shaded areas on the graphs.

eFigure 9 – please indicate the meaning of the shaded areas on the Kaplan-Meier plots (blue, orange and grey).

eFigure 12 – please define the meaning of the grey dashed line for the reader.

eTables – as above, suggest ‘[n, (%)]’ for clarity, throughout, please clearly define numerical values for the reader (i.e. those within and outside of parentheses).

eTable 11 – please clearly define the numerical values contained within parentheses for the reader. Please also indicate the meaning of the shaded area of the table.

eTable 12 & 13 – as above. ? CIs? If so, please separate upper and lower bounds with commas as opposed to hyphens.

Supporting References – please see above and format as detailed. Please ensure all web references include an access date.

Comments from the reviewers:

Reviewer #1: "Assessing eligibility for lung cancer screening: Parsimonious multi-country ensemble machine learning models for lung cancer prediction" describes the development and validation of an ensemble machine learning model with the AutoPrognosis open-source software, with the training data taken from UK Biobank and the control arm of the US National Lung Screening randomized control trial, and the external validation data taken from the US Prostate, Lung, Colorectal and Ovarian Screening Trial (PLCO). The claim of parsimony is based on the use of just three common input risk factors (age, smoking duration, pack-years of smoking), as compared to the up to 19 risk factors required by other existing risk models. It is claimed that the parsimonious ensemble models could attain similar results as the less-parsimonious existing risk models, both for predicting five-year risk of death from lung cancer, and five-year risk of lung cancer incidence.

The simplification of risk models for lung cancer has the potential to be greatly impactful, especially in regions where the collection and recording of relevant clinical variables is less developed. However, there remain a number of issues and concerns that might be addressed:

1. While generalisability to multiple settings and the possibly lower applicability of models validated on only US datasets to global healthcare systems are stated as motivations for this study, this study appears to also be (externally) validated on only a single US dataset (PLCO). As such, the justification for a claim of generalisability might be further justified, ideally with validation on more-diverse datasets if possible.

2. The use of only subjects from the control arm of the US National Lung Screening randomized controlled trial for model development might be briefly discussed. In particular, would the treatment(s) in the non-control arm be expected to be widespread? If so, would the presence (or absence) of such treatment be a relevant input to any risk model?

3. In the Introduction section, it is stated that "Screening for lung cancer... has been associated with a 20-24% reduction in lung cancer-specific mortality amongst those at high risk". The duration over which this mortality is applicable might also be included.

4. In the Introduction section, it is stated that "identifying individuals for lung cancer screening based on risk prediction models has been shown to have both better benefit-to-harm profiles and cost-effectiveness than using risk factors alone". The distinction between "risk factor-based (strategies/models)" and "risk prediction models", might be briefly explicated, since the distinction might not be immediately clear to readers (as risk factor-based strategies appear to also involve functions of multiple risk factors)

5. In the Data sources and study population subsection, it is stated that 216,714 subjects were included from UK Biobank, and then 26,616 subjects from US NLST. Was any particular balancing/weighing of the data from the two datasets performed?

6. In the Outcomes subsection, it is stated that lung cancer diagnosis and primary cause of death were confirmed by review of medical records and death certificates. The expected coverage on the two datasets might be clarified, if possible.

7. In the Model development subsection, the ensemble is stated to consist of four pipelines, for both models. It is stated in the Appendix that ensembles of up to four different pipelines were considered. It might be clarified whether the upper limit of four was selected due to computational concerns, since both final ensembles containing four pipelines suggest possibly improved performance from additional pipelines.

8. In the Variable selection subsection, it is stated that candidate predictors common to the UK Biobank, NLST and PLCO datasets were considered. The full list of such predictors might be included, possibly in the Appendix.

9. It is then stated that the final list of (three) predictors was selected based on the literature, domain expertise, variable distributions etc, which appears to imply that these three predictors were not fully quantitatively assessed. It might be considered to train a ML model(s) with all candidate predictors, to assess their relative importance quantitatively, possibly through SHAP or similar common feature selection methods.

10. In the Code and model availability subsection, the provided Github link (https://github.com/callta/lung_cancer_risk_models) does not appear valid currently.

11. Ideally, the upper limit of predictive performance with all possible candidate variables (possibly as with the "Full Models" in the Appendix) with the same AutoPrognosis approach might also be discussed, to put the trade-off for a parsimonious model in context.

12. Related to the above, it might be considered to benchmark the existing risk prediction models with only the three selected risk factors while imputing the remaining variables with median/modal values, if appropriate.

13. In the Model development section of the Appendix, it is stated that "All pipelines and ensembles were trained using five-fold cross-validation to maximise the area under the receiver operating curve (AUC), with the highest performing ensemble selected". It might be clarified whether a final ensemble is trained using the hyperparameters from this highest performing ensemble on the full training data, to evaluate the external validation data.

Reviewer #2: This is an interesting and novel application of machine learning to developing cancer prediction models. This work is careful and thorough and enjoyable to read, and an important contribution to the lung cancer modeling literature. They find that only 3 factors (age, pack-years, duration) can yield AUC around 0.8 with good calibration in external validation, although they note that Cox models with those same 3 factors (etable 10) are nearly as predictive as the machine learning models. This suggests that existing models require too much information and this type of model could be more productively used to more easily identify a large population who might be interested in screening, as a prelude to clinical visits that assess more risk factors and personal preferences to decide on entering screening.

I think etable 10 is so important, it should be promoted to the main paper. It would help convince the reader that 3 factors really do suffice for good overall predictive performance.

Although the authors' point is well taken, a drawback of leaving out other risk factors could be poor performance in those subgroups denoted by left-out factors such as family history, BMI, race, sex, etc. To see if these new models are well-calibrated and have good AUC for those groups, it would be important to extend table 2 on AUC to include all factors left out but still in LCDRAT or M2012. Then, in addition, also produce a table of the E/O ratios (and their CIs) for each factor in LCDRAT or M2012 (age, race, sex, family Hx, BMI, COPD, etc.). I expect the new UCL models will underestimate risk in people with family history, low BMI, african-americans, those with COPD/emphysema, etc. and that would be an important caveat. The lesson could be, while 3 factors suffice to have great overall performance, including more factors would improve performance within subgroups, and that might be important for purposes of equity and fairness to subgroups. I think this table would be very important.

You note that both your models and existing models overestimate risk in UK Biobank. But UK biobank is composed of healthy volunteers who have half the mortality risk of the UK general population. This suggests that being well-calibrated to a cohort may imply underestimating risk in the general population, especially for minority subgroups that are underrepresented in research cohorts. Can you comment on implications of this for calibration and prediction in the general population? Also can you comment how your model could be used in another country - it might be poorly calibrated and then how would you recalibrate it?

I could not access the github site for the models - can you verify it is up?

Reviewer #3: My review Comments to Authors are attached in a PDF and Word files. The reason for this is that a screen shot of the ensemble model web link is included and it is uncertain whether it transports well outside of the PDF version. 

Reviewer #4: This is an important and well done study. The aim of the study is to use ensemble machine learning with minimal inputs to predict incidence and death from lung cancer in a screened population. The goal was better understand the trade-offs between an individualized model with multiple inputs (PLCOM2012 for example) and capturing as many people as possible who are eligible for screening. The model was developed and validated in multiple settings with large numbers of patients. The methods are sound. I have a few suggestions which may help improve this manuscript. They are mainly aimed at putting the work into clinical context for the reader. The important aspect of this study is the need for entering only three variables that are easily obtainable. 

specific comments:

Introduction: In the introduction you state that it would take 87 full time employees a year to identify 1 million screen eligible individuals. I don't mind you stating that but then somewhere else in the paper you need to do that calculation for what it would take to install and use your model for the same purpose.

Results/Discussion: I think it is important to provide more granular examples of the performance of the model within certain subgroups rather than pointing just to the tables in the text and supplement. For, example the USPSTF 2021 guidance expanded from age 55 to 50. How does the model work in this age group. Your models go to age 40. Do you recommend considering screening elibible patients from 40 on up given that the model appears to work well in the 40-49 age group. How would you see this being implemented? An on-line tool that inputs the 3 variables and a risk score is provided. What would be the risk threshold that meets the eligibility for screening compared to what is currently being utilized. This type of clinical context is needed.

[LINK]

---

## [Decision Letter · Decision Letter 2]

6 Jul 2023

Dear Dr. Callender,

Thank you very much for submitting your manuscript "Assessing eligibility for lung cancer screening:

Parsimonious multi-country ensemble machine learning models for lung cancer prediction" (PMEDICINE-D-23-00694R2) for consideration at PLOS Medicine. 

Your paper was evaluated by a senior editor and discussed among all the editors here and sent again to independent reviewers, including a statistical reviewer. The reviews are appended at the bottom of this email and any accompanying reviewer attachments can be seen via the link below:

[LINK]

In light of these reviews, I am afraid that we will not be able to accept the manuscript for publication in the journal in its current form, but we would like to consider a revised version that addresses the reviewers' and editors' comments. Obviously we cannot make any decision about publication until we have seen the revised manuscript and your response, and we plan to seek re-review by one or more of the reviewers. 

We expect to receive your revised manuscript by Jul 27 2023 11:59PM. Please email us (plosmedicine@plos.org) if you have any questions or concerns.

We look forward to receiving your revised manuscript. 

Sincerely,

Philippa Dodd, MBBS MRCP PhD

PLOS Medicine

plosmedicine.org

GENERAL

Thank you for your detailed and considered responses to previous editor and reviewer comments. Please see below for further comments which we require that you address in full.

Please cite your Supporting Information as outlined here: https://journals.plos.org/plosmedicine/s/supporting-information

* Considering that the comments from the statistical and subject reviewer (please see below) pertain to further clarifications of your model and analyses, and in view of the fact that revisions have the potential to alter the data, it is for this reason that we have invited a further major revision. *

DATA AVAILABILITY STATEMENT

The Data Availability Statement (DAS) requires revision. For each data source used in your study: 

TITLE

The title is quite long, please revise. Your title must be nondeclarative and not a question. It should begin with main concept if possible. "Effect of" should be used only if causality can be inferred, i.e., for an RCT. Please place the study design ("A randomized controlled trial," "A retrospective study," "A modelling study," etc.) in the subtitle (ie, after a colon). 

Suggest - ‘Assessing eligibility for lung cancer screening using parsimonious multi-country ensemble models: A machine-learning study’ or similar. 

AUTHOR SUMMARY

Line 67 – suggest, ‘Combining predictive performance with administrative simplicity could allow for streamlined delivery of risk-tailored lung cancer screening.’ This statement is a little vague suggest elaborating. How does the simplified approach streamline things – reaches regions with limited availability of patient data, for example.

Line 69 – suggest, ‘Further research should focus on the application of this model design to other conditions such as…’

In the final bullet point of ‘What Do These Findings Mean?’, please describe the main limitations of the study in non-technical language.

INTRODUCTION 

Line 110-111 – suggest ‘per year’ Is the obstacle you refer to cost? Please clarify

Line 148 – ‘(PLCO) [28] trial’ please add a space between the sets of brackets

METHODS and RESULTS

Line 185 – ‘(SHAP) [33]’ please add a space between the sets of brackets 

Line 189 – please indicate where in the appendix that the details you refer to can be found

DISCUSSION

Thank you for structuring your discussion as suggested. Please expand on the limitations of your study which are rather brief.

We could not find a section which details the strengths of your study, please include.

Lines 366 – 374 – suggest moving this text to the second paragraph it seems out of place here (and the opening sentence a bit repetitive this far through the discussion. Please also remove the phrase ‘state-of the-art’

Line 386 – please remove ‘In summary’

REFERENCES

Please ensure that all web references listed include an access date, ref #7 and #31-34, for example

Is ref #29 a complete reference?

SUPPORTING INFORMATION

eFig7 – please define SHAP in the caption/footnote

Please ensure that the required referencing format is applied to the supporting files. 

Please list up to but no more than 6 author names followed by et al. 

Please ensure that journal name abbreviations are those found in the National Center for Biotechnology Information (NCBI) databases. 

Please ensure that all web references include an access date

Please see our website for other reference guidelines https://journals.plos.org/plosmedicine/s/submission-guidelines#loc-references

SOCIAL MEDIA

If not already done so, to help us extend the reach of your research, please detail any Twitter handles you wish to be included when we tweet this paper (including your own, your coauthors’, your institution, funder, or lab) in the manuscript submission form when you re-submit the manuscript.

Comments from the reviewers:

Reviewer #1: We thank the authors for substantively addressing our previous comments, and the emphasis on recalibration for future application in the responses to other reviews. A couple of points might yet be considered:

1. In the response to the original Point 5, it is clarified that the 216,714 UK Biobank and 26,616 US NLST subjects were combined without any weighting. However, the additional value of including the US NLST data is then not immediately clear, as it might be dominated by the UK Biobank data.

It might thus be considered to perform brief ablation experiments with only Biobank/only NLST data, to quantify the added value of including additional data sources.

2. Moreover, for the NLST data, the background does not appear to be analyzed as thoroughly as for Biobank data (as in the "Variable recoding, missing data, and multiple imputation" section of the supplementary material). In particular, it is not immediately clear as to the number of subjects progressing to/dying from lung cancer, as clearly laid out for Biobank data in eTable1. It might thus be considered to present the NLST data in more detail too.

Reviewer #2: none

Reviewer #3: All in all, the authors responded well to the comments made by reviewers. I only have minor comments left. 

Abstract

"Ensemble machine learning could support the development of highly parsimonious prediction models that maintain the performance of more complex models whilst maximising simplicity and generalizability" 

Regarding "maintain the performance of more complex models whilst maximising simplicity and generalizability", 

it is difficult to conceptually appreciate that a model that looks at only a limited number of risk factors can be as robust at estimating personal risk as a model that has a wide diverse collection of predictors, which taken together as a package summarize a person's global risk more comprehensively than focussing on a narrow list that consists of three of predictors age and smoking. 

The eFig 7 shows the relative importance contribution of predictors in the full model. It shows that combined BMI, quit-years, smoking intensity, family history of lung cancer and COPD, contribute substantially more to prediction than do pack-years or age. Thus, they may be meaningful in some setting. 

The model development and validation came from high quality and complete data. When this model is applied to electronic medical record data in which pack-years and duration smoked is incomplete and out of date, then ancillary data from other variables can be valuable in augmenting age/smoking predictors, for example, BMI, history of COPD, family history of lung cancer, personal history of cancer, race, are predictive variable which often at least in part are available in patients EMR which can offset absence or measurement error in smoking data. Thus, it is expected that a fuller model will be more robust than a parsimonious bare-bones model. 

The authors describe and explain that their use of model would involve collection of accurate data, not reliance on electronic medical records, which makes my comments regarding EMR less relevant in their setting. 

"We externally. validated our models amongst 49,593 participants in the chest radiography arm and all

80,659 ever-smoking participants in the US Prostate, Lung, Colorectal and Ovarian Screening Trial (PLCO)." 

In the Abstract, the thinking behind this strategy is unclear for readers. 

"our prognostic models would be trained on a wider range of participants, with the potentially improved model performance" 

Although the term "prognostic model" has been widely and loosely used in recent years, I prefer to use "prognostic" to describe survival in individuals with cancer (individuals with disease) and not for prediction models describing risk of developing lung cancer in individuals without disease who will in the future develop disease. 

From response to reviewer #3. 

"As the difference in discrimination between the full models and the three-variable models was not statistically significant, we took forward the three-variable models." 

Was a p-value or confidence intervals for the difference presented?

eTable 14, Overall calibration lower confidence interval for LCRAT is missing second decimal place. Missing second decimal occurs elsewhere too. 

The eTable 14 indicates that the UCL-D and UCL-I are not well calibrated, with 47% and 35% excess risk estimates in the UCL-D and UCL-I models respectively. That the final pooled estimated E/R is much closer to 1 indicates that the calibration in the non-UK Biobank data may be poorly calibrated in the opposite direction, so that they cancel out. 

In a few places where we could readily check the results, the numbers did not match. In particular, the AUC and E/O for PLCOm2012 in the PLCO intervention arm and entire sample were underestimated. Do the authors think that this was due to computing issues, program used to prepare estimates or modifications of dataset? Examples are provided:

Table 4: Calibration overall and by subgroup in the PLCO chest radiography cohort

Tabel 4 reports Ratio of expected-to-observed cancers with 95% confidence intervals overall for PLCOm2012 to be "0.93 (0.86, 0.99)"

Our STATA results in the PLCO intervention arm are different, with E/O = 0.95 and AUC 0.797.

STATA result: 

/* . brier lcancer6yr PxbMFP if H_rndgroup==1 

Mean probability of outcome 0.0181

 of forecast 0.0172

display 0.0172 / 0.0181 // .95027624 E/0 = 0.95

ROC area 0.7967 p = 0.0000 */ 

eTable 16: Discriminative accuracy (AUC) in the whole PLCO cohort

eTable 16 reports that in the PLCO overall for PLCOm2012 the AUC (95% CIs) are "0.79 (0.78, 0.8)". 

When we calculate them with original data and model, the results are 0.80 (.79 to 0.81). 

STATA printout

/* . roctab lcancer6yr PxbMFP if H_rndgroup<2

 ROC Asymptotic normal 

 Obs area Std. err. [95% conf. interval]

 74,218 0.7995 0.0056 0.78859 0.81038 

REPORTED IN PLOSmed manuscript 0.79 (0.78, 0.8) 

OUR STATA RESULTS ARE 0.80 (.79 TO 0.81) */

[LINK]

---

## [Decision Letter · Decision Letter 3]

16 Aug 2023

Dear Dr. Callender,

Thank you very much for re-submitting your manuscript "Assessing eligibility for lung cancer screening using parsimonious ensemble machine learning models: A development and validation study" (PMEDICINE-D-23-00694R3) for review by PLOS Medicine.

I have discussed the paper with my colleagues and it was also seen again by two reviewers. I am pleased to say that provided the remaining editorial and production issues are dealt with we are planning to accept the paper for publication in the journal.

[LINK]

We look forward to receiving the revised manuscript by Aug 23 2023 11:59PM.   

Sincerely,

Philippa Dodd, MBBS MRCP PhD

Senior Editor 

PLOS Medicine

plosmedicine.org

Requests from Editors:

GENERAL

Thank you for your previous responses to editor and reviewer comments. Please see below for further comments which we require you address in full.

AUTHOR SUMMARY

Line 54 – suggest ‘existing approaches are…’.

Line 69 – this is repetitive of the statement at line 61 suggest removing this statement (see ** below also).

Line 71 – please replace ‘our’ with ‘this’. 

** when considering the points detailed at lines 69 and 71, please include the implications of your study findings. How does this help us and who does it help? What are the (potential) implications for lung cancer outcomes. Please be explicit but brief. **

TABLES

Tables 1, 2, 3, 4 – please include definitions of the abbreviations used for the different models (and datasets) in the footnotes (as for table 3). For example, LCRAT – lung cancer incidence, LCDRAT – lung cancer death (as we understand things). As well as ‘UCL’. It looks as though this task has been started but not completed. Please amend throughout including the supporting information where relevant.

Table 4 – please remove the repeated table header where the table crosses the page.

FIGURES

As above please define the abbreviations used for the models (and datasets) in all figures.

Figure 2 caption – previously you detail LLPv2 but her you detail ‘LLP’ please amend for consistency of reporting (and throughout).

REFERENCES

For web references please detail an ‘accessed’ date as opposed to a ‘cited’ date.

SUPPORTING INFORMATION

When citing your supporting information please place S1 before the table/figure/text you describe. For example, ‘S1 Fig’ as opposed to ‘Fig S1’. Please check and amend throughout including in the supporting information itself.

Appendix - Please list the appendix and ‘S1 Appendix’.

Appendix page 7 – please define NLST and PLCO.

Please see here for further information https://journals.plos.org/plosmedicine/s/supporting-information

Supporting information captions - In the published article, supporting information files are accessed only through a hyperlink attached to the captions. For this reason, you must list captions at the end of your manuscript file. You may include a caption within the supporting information file itself, as long as that caption is also provided in the manuscript file. Do not submit a separate caption file.

Tables – as for the main manuscript please define all abbreviations/acronyms used for the models.

S5 Table – please define NLST. Please remove the repeated table title where the table splits across the page.

S6 Table - please define PLCO. Please remove the repeated table title where the table splits across the page.

S7 Table – please define PLCO. Please remove the repeated table title where the table splits across the page.

S8 Table – please define PLCO and NLST.

S9 Table – please define UCL.

S10 -S18 Table – please define all abbreviations/acronyms used for the models in each.

Figures – as for the main manuscript please define all abbreviations/acronyms used for the models in each of the figures.

References – please ensure reference format is consistent with that for the main manuscript. Please see here for further details https://journals.plos.org/plosmedicine/s/submission-guidelines#loc-references

Comments from Reviewers:

Reviewer #1: We thank the authors for addressing our remaining comments. It might then be considered to briefly mention that analysis had indeed been performed with and without the NLST data, and that the NLST data was included for calibration and to add diversity.

Reviewer #3: Discussion: "Our analyses have been performed in research cohorts rather than rouFnely collected

electronic health records that may beIer the broader populaFon."

I believe that this wording is grammatically incorrect. 

I have reviewed responses made to editor and all reviewers and am satisfied with the authors responses. I have nothing additional to add.

[LINK]

---

## [Editor Report · Decision Letter 4]

29 Aug 2023

Dear Dr Callender, 

On behalf of my colleagues and the Academic Editor, Dr. Aadel Chaudhuri, I am pleased to inform you that we have agreed to publish your manuscript "Assessing eligibility for lung cancer screening using parsimonious ensemble machine learning models: A development and validation study" (PMEDICINE-D-23-00694R4) in PLOS Medicine.

Prior to publication we require that you make the following revision:

* Line 255 – code and model and availability. Please include your statement (as detailed below) and the relevant links in the manuscript submission form under 'Data Availability' and remove from the main manuscript. It will be compiled as metadata at the time of publication.

‘To facilitate use of the UCL models, we have developed a website and have made the models themselves available: https://github.com/callta/lung-cancer-models. The underlying code for AutoPrognosis is available from https://github.com/vanderschaarlab/AutoPrognosis.’

PRESS

Best wishes, 

Philippa Dodd, MBBS MRCP PhD 

PLOS Medicine